# UPF1-like helicase grip on nucleic acids dictates processivity

Joanne Kanaan[1], Saurabh Raj[1,2,4], Laurence Decourty[3], Cosmin Saveanu [3], Vincent Croquette[1,2] & Hervé Le Hir[1]

Helicases are molecular engines which translocate along nucleic acids (NA) to unwind double-strands or remodel NA–protein complexes. While they have an essential role in genome structure and expression, the rules dictating their processivity remain elusive. Here, we developed single-molecule methods to investigate helicase binding lifetime on DNA. We found that UPF1, a highly processive helicase central to nonsense-mediated mRNA decay (NMD), tightly holds onto NA, allowing long lasting action. Conversely, the structurally similar IGHMBP2 helicase has a short residence time. UPF1 mutants with variable grip on DNA show that grip tightness dictates helicase residence time and processivity. In addition, we discovered via functional studies that a decrease in UPF1 grip impairs NMD efficiency in vivo. Finally, we propose a three-state model with bound, sliding and unbound molecular clips, that can accurately predict the modulation of helicase processivity.

[1] Institut de biologie de l'Ecole normale supérieure (IBENS), Ecole normale supérieure, CNRS, INSERM, PSL Research University, 46 rue d'Ulm, 75005 Paris, France. [2] Laboratoire de Physique Statistique, École Normale Supérieure, PSL Research University, Université Paris Diderot Sorbonne Paris-Cité, Sorbonne Universités UPMC Université Paris 06, CNRS, 24 rue Lhomond, 75005 Paris, France. [3] Génétique des Interactions Macromoléculaires, Genomes and Genetics Department, Institut Pasteur, 25-28 rue du docteur Roux, 75015 Paris, France. [4] Present address: Molecular Biophysics Group, Peter Debye Institute for Soft Matter Physics, Universität Leipzig, Linnéstraße 5, 04103 Leipzig, Germany. These authors contributed equally: Joanne Kanaan, Saurabh Raj. Correspondence and requests for materials should be addressed to V.C. (email: vincent.croquette@lps.ens.fr) or to H.L.H. (email: lehir@ens.fr)

Helicases are ubiquitous molecular motors found in all living organisms. These enzymes are organized around a core domain that simultaneously binds nucleotide triphosphates (NTPs) and nucleic acids (NA) in a sequence-independent manner, and converts the chemical energy of NTP hydrolysis into mechanical activities on NA. Hence, helicases can lock onto NA, translocate on single-strands (ss) or double-strands (ds), and apply a mechanical force either to unwind dsNA or to remodel NA–protein complexes[1–3]. This versatile ability to act on NAs justifies the ubiquitous involvement of helicases in every NA-related process including DNA replication, repair, recombination, transcription, and every event of post-transcriptional gene regulation[2,4,5]. Despite seemingly comparable cores, helicases intervene in precise processes and sometimes act on specific NA substrates. The specificity of their actions is conferred by accessory domains either flanking or inserted in the helicase core. These additional domains can carry complementary catalytic activities, affinity for specific NA sequences, or ensure interactions with protein partners, as most helicases are part of protein complexes[6–8].

Helicase coding genes have been classified into six superfamilies (SF1-SF6) and subsequent subfamilies. SF1 and SF2 are the largest groups of RNA and DNA helicases generally acting as monomers or dimers while SF3–SF6 encompass multimeric ring-shaped helicases[7,9]. The helicase domain of SF1 and SF2 enzymes is composed of conserved and characteristic motifs[2,7,10,11]. However, the presence of signature motifs or the affiliation of a helicase to a specific family cannot foretell its intrinsic biophysical properties and physiological functions. The UPF1-like family is a good example of functional diversity occurring between sister helicases. This family is formed of a group of 11 enzymes belonging to the SF1-B 5′–3′ helicases[10]. Among them one finds IGHMBP2 (immunoglobulin helicase mu-binding protein 2), a helicase related to mRNA translation and responsible for distal spinal muscular atrophy with respiratory distress type 1[12,13], SetX/Sen1 (Senataxin), which is involved in transcription termination, R-loop resolving and is linked to amyotrophic lateral sclerosis[14], and MOV10 (Moloney leukemia virus 10), which is involved in miRNA-dependent regulation[15,16].

Members of UPF1-like helicases present a common helicase domain organization with two RecA-like domains (1A and 2A) containing conserved helicase motifs and two domains (1B and 1C) protruding from the first RecA domain 1A[10,17–20]. The prototype, UPF1 (Up-frameshift 1) is a multifunctional RNA and DNA helicase. It is implicated in telomere maintenance and telomerase activity regulation[21] and various mRNA decay pathways[22–24]. UPF1 is best-known for its essential role in nonsense-mediated mRNA decay (NMD), an eukaryotic surveillance pathway that eliminates aberrant messenger RNAs (mRNAs) carrying premature termination codons (PTC) and modulates the expression level of normal mRNAs-presenting NMD substrate features[25,26].

We recently assessed some of the biophysical properties of UPF1 by using magnetic tweezers to manipulate DNA and RNA hairpins at the single-molecule scale[27]. We monitored the activity of the helicase domain of human UPF1 (hUPF1) and discovered some unexpected properties. As a monomer, hUPF1 was able to unwind long dsNA and translocate onto ssNA with a processivity exceeding 10 kilobases (kb). Though the processivity of other UPF1-like helicases has not been determined, the processivity of hUPF1 exceeds that of the DNA helicases UvrD or Rep in their monomeric state when measured by single-molecule approaches[28,29]. The processivity of UPF1 is particularly surprising given that it translocates onto NA as a monomeric unit at a rate at least one order of magnitude slower than UvrD, Rep, or the SF2 RNA helicase NS3[27,29–32]. Therefore, to cover similar

distances, hUPF1 must stay a much longer time on its substrate. In addition, the progression of UPF1 onto RNA or DNA is not affected by NA-bound proteins[27]. These observations raise the question of whether these peculiar attributes are specific to hUPF1 or whether they are shared with the related UPF1-like helicases.

In the present study, we show that the high processivity of hUPF1 is shared with its homolog from the yeast Saccharomyces cerevisiae (yUPF1) but not with the very similar human helicase IGHMBP2. Structural comparison between core helicase domains of both proteins led to the design of a series of mutants differentially affecting UPF1 processivity. Using a single-molecule binding assay, we measured the strength of UPF1 grip on NA and established a correlation between helicase grip, binding lifetime, and the duration of translocation. In addition, the study of a mutant affecting UPF1 grip in yeast demonstrates its relevance for UPF1 function in NMD.

## Results

**Processivity is variable amid UPF1-like helicases.** We first determined whether the helicase core of yeast UPF1 (yUPF1) is as processive as the helicase core of human UPF1 (hUPF1). Indeed, both proteins are key NMD factors, and both core domains present high sequence and structural similarities (Supplementary Figure 1 [33]). Using a magnetic tweezers setup, we manipulated single 1.2 kb DNA hairpins tethering super-paramagnetic beads to a glass surface. Using video-microscopy, we tracked the beads' positions to monitor changes in the extension of the hairpins over time (Fig. 1a [27,34]), all the while applying a controlled force on the hairpin fork using magnets to pull on the tethered beads. yUPF1 (in an appropriate buffer with ATP) was injected in this setup at the lowest possible concentration to observe monomeric single-molecule events. Under a constant tension of 7 pN and at saturating ATP concentration (2 mM), yUPF1 molecules generated characteristic saw-tooth tracks (Fig. 1b) similar to hUPF1 (Supplementary Figure 2A). The rising and falling edges correspond, respectively, to complete hairpin unwinding (Fig. 1a, steps 2–3) and rezipping in the wake of the enzyme following translocation on single strand (ss) (Fig. 1a, step 4). yUPF1 unwinding and ss translocation rates were significantly higher (10.2 and 13.4 bp s$^{-1}$, respectively) than those of hUPF1 (0.32 and 1.92 bp s$^{-1}$) consistent with their different ATP consumption rates (Supplementary Figure 2B). To assess yUPF1 processivity, we first calculated the probability of fully unwinding the 1.2 kb DNA hairpin using 58 events that initiated on fully closed hairpins. Using this probability, we estimated the unwinding processivity to be superior to 10 kb ($n = 58$) (Supplementary Figure 2C). We thus concluded that a high processivity is a conserved UPF1 feature shared between yeast and human homologs[33].

As UPF1 is the prototype of the UPF1-like helicase family, we broadened our scope and examined whether high processivity is also a feature of other UPF1-like helicases. To this end, we selected the human IGHMPB2 protein, a helicase linked to DSMA1 respiratory disease[12,13]. Guenther et al.[35] previously described the ATP-dependent 5′–3′ helicase activity of a recombinant full-length IGHMPB2 on small RNA and DNA duplexes. However, no unwinding assays on the human IGHMBP2 helicase domain alone have been done so far. Using our single-molecule conditions, we assayed the isolated helicase core of IGHMBP2, which did not display any observable unwinding activity (Fig. 1c) despite a detectable (albeit sevenfold lower) ATPase activity in vitro (Supplementary Figure 2B). However, when flanked by its C-terminal domain (Supplementary Figure 2D) which increases its binding affinity[19], IGHMBP2 presented short unwinding events with a small processivity

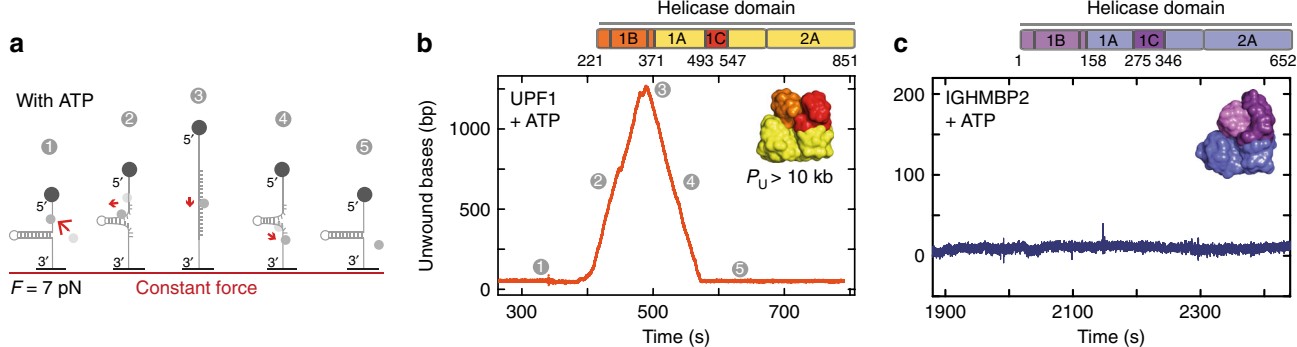

**Fig. 1** Closely related SF1-B helicases present very different processivities. **a** Schematic representation of the single-molecule experimental setup. A 1.2 kb DNA hairpin substrate tethers a magnetic bead to a glass surface and is subjected to the tension exerted by magnets pulling on the bead. Different phases of beads movements in the presence of ATP and an active 5′ to 3′ unwinding helicase are shown not to scale. **b** Helicase domain organization of yUPF1 made of the RecA domains 1A and 2A (yellow) and the domains 1B (orange) and 1C (red) protruding from the domain 1A (Protein Data Bank (PDB) identifier 2XZP). Experimental trace showing unwound bases with 2 mM ATP and yUPF1 at a constant tension on the hairpin of 7 pN. **c** Same as **b** with IGHMBP2 except that domains 1A and 2A are in blue, 1B in light purple, and 1C in purple (PDB 4B3F)

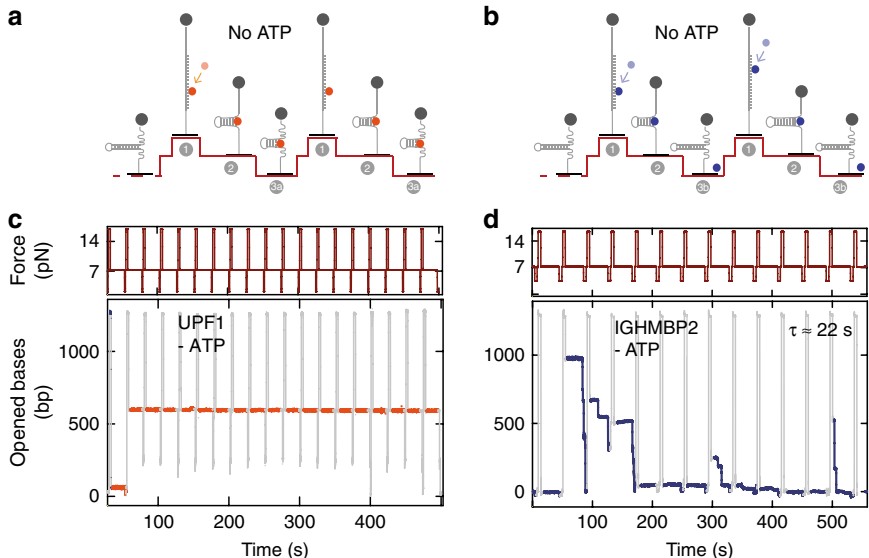

**Fig. 2** Single-molecule binding assay (SMBA) to assess helicase binding. **a**, **b** Schematic representation of the single-molecule binding assay cycle in absence of ATP. DNA hairpin is initially completely closed. Step 1, a force >15 pN is applied to fully unzip the DNA hairpin which is kept in an open state for 4 s to allow a helicase in solution to bind to the ssDNA. Step 2, force reduction to 7 pN refolds hairpin to its natural state, unless a helicase is bound. Force is held at 7 pN for 20 s leading to helicase residence on substrate over several cycles (3a) or expulsion after one or a few cycles (3b). Step 3, force is reduced <3 pN to check that the hairpin can still completely close. The whole process is repeated several times. **c** Upf1 remains bound to the substrate at its initial position and blocks hairpin closure over all force cycles. **d** IGHMBP2 transiently binds then falls off the substrate after one or a few cycles. Values on $y$-axis in **c** and **d** indicate hairpin opening during step 2 of the binding assay described in **a** and **b**

estimated to be 19 bp only (Supplementary Figure 2D, E; unwinding speed 5 bp s$^{-1}$). Taken together, our experiments reveal that high processivity is not shared uniformly among the UPF1-like helicase family, in spite of similar core domains.

**Binding lifetime varies between sister helicases**. How to explain such a diverse behavior between closely related helicases? To travel over a long distance at slow speed, a helicase must remain bound to its NA substrate despite conformational changes occurring with every ATP hydrolysis cycle (Supplementary Figure 3A). In other words, a helicase with a short binding lifetime and a slow speed will only travel a few base pairs before falling off its substrate. Hence, to understand the differences between yUPF1 (UPF1 for simplicity hereafter) and IGHMBP2, we first compared their binding lifetime on a substrate. In bulk assays,

both proteins bind NA in absence of ATP[35,36], so we developed an ATP-free single-molecule binding assay (SMBA) to measure their residence time ($\tau_R$) in a stationary state (Fig. 2a, b). In contrast to bulk assays, SMBA detects dissociation events and therefore measures the binding off rate. In each experiment, we injected the helicase in a buffer lacking ATP, then performed a series of cycles with 3-phases. During the first phase DNA hairpins are fully unzipped for 4 s at a force >15 pN, allowing random helicase binding to ssDNA (Fig. 2a, step 1). In the second phase, the force is reduced to 7 pN, and hairpins start refolding to their natural closed state, unless a helicase acts as a roadblock (Fig. 2a, step 2). While the force is held at 7 pN for 20–30 s, the closing hairpin fork constantly pushes on the bound helicase and challenges its binding. Thus, the helicase either remains on its substrate (Fig. 2a, step 3a) or is ejected (Fig. 2a, step 3b). A third phase is performed where force is reduced to 3 pN to verify

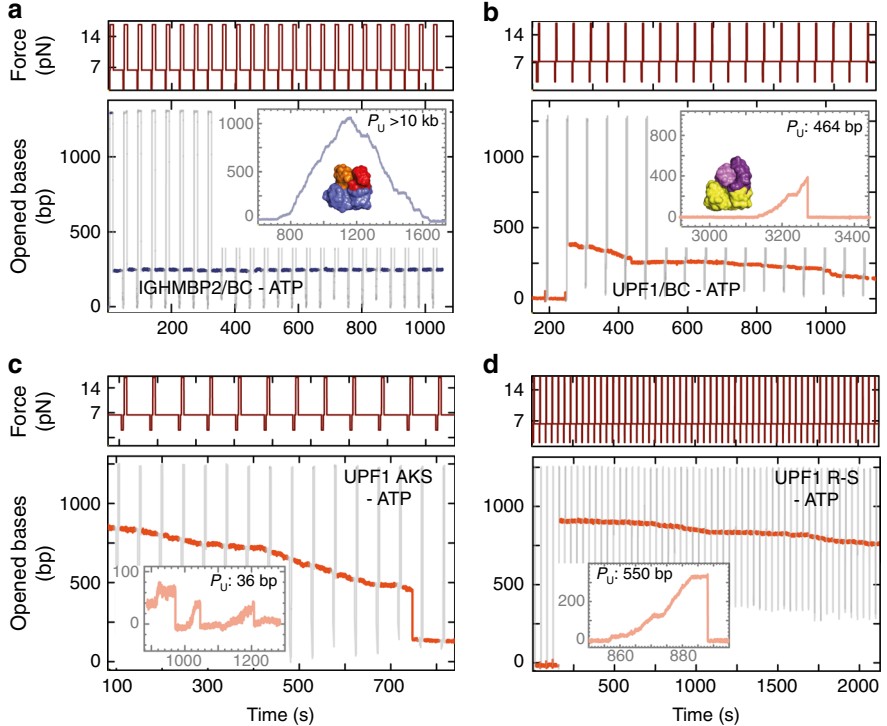

**Fig. 3** Single-molecule characterization of binding grip and its impact on processivity. **a**, **b** Effect of protrusion swapping on IGHMBP2-BC (**a**) and UPF1-BC (**b**) processivity in the presence of ATP (inside frames) and binding strength in absence of ATP (-ATP). Unwinding processivity ($P_U$) is indicated. Protrusion swapping leads to gain of IGHMBP2 processive unwinding and stable binding (**a**), and a reduction of UPF1 processivity and grip on substrate (**b**). **c**, **d** Same as **a** with the mutants UPF1 AKS (**c**) and UPF1 R-S (**d**). Mutating a previously uncharacterized UPF1 loop reduces processivity and binding grip

complete hairpin closure. This force cycle is repeated multiple times over hundreds of seconds. During this test, single-UPF1 molecules attached and remained at their initial binding positions until recording was stopped, despite all opposing forces ($n = 11$) (Fig. 2c). UPF1 residence time was estimated to be larger than 5500 s (Supplementary Table 1). Under similar conditions, whether with or without its C-terminal domain, IGHMBP2 poorly resisted to the closing fork and was ejected after one or two cycles (Fig. 2d, Supplementary Figure 3B). IGHMBP2 residence times followed exponential distributions with a mean of 20 sec (no C-ter, $n = 48$) (Supplementary Figure 3C) or 60 s (with C-ter, $n = 27$). These results strongly suggested a tight link between residence time and unwinding processivity, which relies on the ability to remain bound to the NA substrate over multiple ATP hydrolysis cycles (Supplementary Figure 3A).

**Structural features grant UPF1 a strong grip on substrates**. The differences in residence time and processivity between UPF1 and IGHMPB2 helicases were intriguing, especially considering the structural similarities between their core domains. Lim et al.[19] previously reported a superposition of IGHMPB2 and UPF1 structures, showing that both helicases share overall similar core motor domains (Supplementary Figure 4A, B). Hence, we looked for specific structural features likely impacting residence time and contributing to binding. Both UPF1 and IGHMBP2 are formed of two RecA-like domains designated as Rec1A and Rec2A. They also share distinctive protruding sub-domains 1B and 1C embedded within Rec1A, in contrast to SF1-A helicases such as Rep, UvrD, and PcrA, which have one protruding domain on each RecA contributing to processivity[30]. Both domains 1B and 1C could impact UPF1 and IGHMBP2 binding differently. Indeed, domain 1B shows different folding and movement upon RNA and/or nucleotide binding in each helicase, and deletion of domain 1C abolishes NA binding in UPF1[17,19,33]. To evaluate their importance,

we simultaneously swapped domains 1B and 1C between UPF1 and IGHMBP2 (UPF1/BC and IGHMPB2/BC) and investigated the resulting chimeric helicases. Remarkably, IGHMBP2/BC mimicked UPF1 behavior as its processivity markedly increased (>10 kb), along with its residence time (9540 s) during SMBA (Fig. 3a). In contrast, the swap negatively affected UPF1/BC residence time on DNA (930 s) and reduced its processivity (464 bp) (Fig. 3b), confirming the impact of these auxiliary domains on the residence time and unwinding processivity.

Interestingly, UPF1/BC also displays a singular binding property. As the force cycles proceeded during SMBA, UPF1/BC progressively slid along the hairpin from its initial binding position to a lower position (Fig. 3b). This sliding reflects a lower resistance to fork pressure and indicates a looser grip on DNA, which might account for the reduction in residence time. We further swapped the 1B or 1C domains, one at a time. IGHMBP2/B and IGHMBP2/C chimeras were insoluble or ATPase inactive, while UPF1 chimeras were active with reduced residence times and unwinding processivities (Supplementary Figure 4C–F; Supplementary Table 1). Taken together, these findings conclusively unravel a crucial role for protrusions 1B and 1C in regulating the binding and unwinding abilities of UPF1-like helicases.

As protrusions 1B and 1C are both attached to Rec1A and interact with the 3′-end of the NA substrates, we further looked in this area for key elements impacting helicase grip and interdomain connection. We identified a small loop at the junction between domains 1A and 1C that undergoes conformational changes upon NA binding[17,33] (Supplementary Figure 5A) and displays sequence variability between UPF1-like helicases (Supplementary Figure 5B). Additionally, in the UPF1 crystal structure, four amino acids (yUPF1 AKSR 484–487) contact NA at the loop level, whereas in the corresponding IGHMBP2 sequence (IGHMBP2, HPAR 267–270) only the arginine contacts NA[19,33]. To study the impact of this loop on UPF1 performance, we generated the

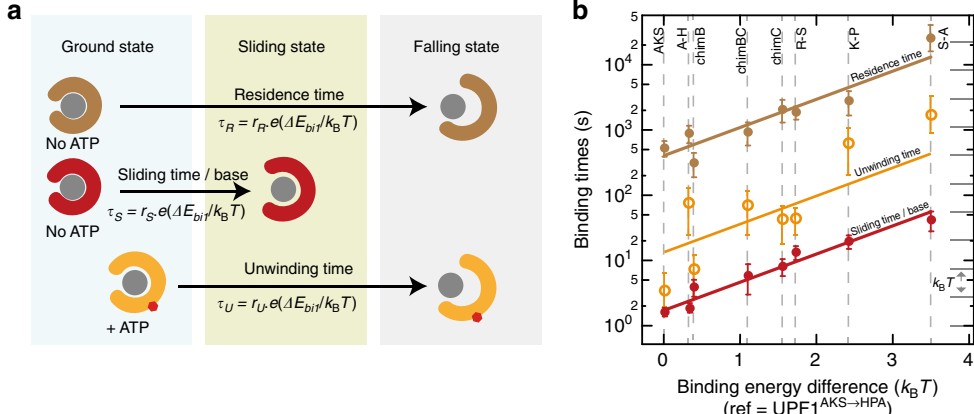

**Fig. 4** Mechanistic model describing helicase grip in different energy states. **a** Schematic representation of the three observable helicase states (see text). Gray circle represents NA. Colored clips represent helicases clamping NA more or less tightly. **b** Binding and sliding times plot versus increasing binding energy differences represented with a logarithmic scale in y. For each helicase $i$, the sliding time $\tau_S$, the unwinding time $\tau_U$ and the falling time $\tau_R$ are drawn in logarithmic scale against the binding energy difference $\Delta E_{bi1} = E_{bi} - E_{b1}$ measured in units of $k_B T$ ($E_{b1}$ corresponds to UPF1$^{AKS \rightarrow HPA}$ binding energy). Each helicase has its three representative points located at the same abscissa. Full lines are exponentials predicted by Supplementary Equation 3 (Supplementary Note). Distance between the lines reflects the energy difference between these three states. Error bars indicate standard deviation

following mutants: UPF1$^{A484 \rightarrow H}$, UPF1$^{K485 \rightarrow P}$, UPF1$^{K485 \rightarrow A}$, UPF1$^{S486 \rightarrow A}$, UPF1$^{R487 \rightarrow S}$, and a triple mutant UPF1$^{AKS \rightarrow HPA}$. Only mutant UPF1$^{S486 \rightarrow A}$ behaved like wild-type UPF1. All other mutants suffered from a reduction in processivity and residence time in SMBA, and displayed a sliding rate that reflects a weaker grip on the substrate (Fig. 3c, d; Supplementary Figure 6A–F). Remarkably, all three parameters vary significantly among the different mutants, showing that small targeted mutations can result in profound behavioral diversity. However, the variation of these parameters was correlated: the mutants with the lower processivities also displayed faster sliding rates and shorter residence times. Altogether, these observations highlight the impact of UPF1 grip on its total residence time whether unwinding or just bound to its substrate.

**A mechanistic model links helicase grip and processivity**. Our study encompasses a large set of functional helicases presenting variable grips on their substrate as well as different binding lifetimes in the absence of ATP and during unwinding. This variety offers the opportunity to rationalize the link between helicase grips and the time they spend on NA. We thus propose a mechanistic model (detailed in the Supplementary Note) in which the helicase behaves as a molecular clip with three different states. In the ground state, the clip is strongly bound to its substrate, in the sliding state the clip is partially opened, and in the unbinding state, the clip is fully opened and falls off (Fig. 4a). To build our model, we measured several parameters gathered in Supplementary Table 1. Using SMBA, we measured for each helicase the average sliding time per nucleotide $\tau_S$ as a marker of grip tightness, and the total residence time $\tau_R$ in absence of ATP. $\tau_S$ and $\tau_R$, respectively, represent the time necessary to shift from the ground state to the sliding state and to the unbinding state. Our data revealed a covariation of $\tau_S$ and $\tau_R$ (Supplementary Figure 7A). These times are set by an Arrhenius law[37], and define how long it takes for thermal fluctuations to reach an energy level sufficient to either slide or fall-off. By fitting our molecular clip model, we calculated the binding energy difference each mutant requires to slide or fall in reference to the enzyme having the smallest energetic needs (UPF1$^{AKS \rightarrow HPA}$). We further plotted the values of the binding energy differences versus the corresponding $\tau_R$ and $\tau_S$ (Fig. 4b). Remarkably, $\tau_R$ and $\tau_S$ almost perfectly follow the model predictions (Fig. 4b) demonstrating that a small alteration in binding energy directly impacts helicase grip and residence

time. Furthermore, while $\tau_R$ and $\tau_S$ widely vary among mutants, the ratio $\tau_R/\tau_S$ is constant, indicating that the transition state to sliding or detaching is the same for all mutants. Similarly, we measured the unwinding time $\tau_U$ in presence of ATP. $\tau_U$ is the time each helicase spent unwinding before falling from its substrate. This time also showed a covariation with $\tau_R$ (Supplementary Figure 7B). Our model reveals that $\tau_U$ correlates with the binding energy (Fig. 4b), showing that the binding lifetime in presence of ATP also depends on the binding energy. Furthermore, $\tau_U$ is always smaller than $\tau_R$ ($\tau_R/\tau_U$ is constant), indicating that helicases are more likely to fall-off during ATP hydrolysis. To test this hypothesis, we chose the moderately affected mutant UPF1$^{R487 \rightarrow S}$ and tested by SMBA the impact of ADPNP, a non-hydrolysable analog of ATP. Interestingly, the addition of ADPNP reduced the binding lifetime of this mutant and steepened its sliding slope (Supplementary Table 1). This result strongly suggests that during a power stroke, the binding energy is reduced due to conformational changes, and a smaller fluctuation is required to reach the opening threshold of detachment. Despite this fragile state, UPF1 manages to keep a tight grip on NA and avoids a detrimental fall while moving thanks to its large binding energy. In contrast, UPF1 mutants suffer from a looser grip and have a higher probability of falling, leading to their reduced processivity. Thus, our model demonstrates that aside from NA composition and fork pressure[31,38], binding energy is a major parameter that determines helicase processivity.

**Loss of UPF1 processivity reduces NMD efficiency**. We wondered if the mechanistic conclusions described above were relevant for UPF1 function in vivo. As UPF1 is an essential NMD factor among all eukaryotes, we evaluated the necessity of its grip and processivity during this quality control pathway. Overexpression of yeast UPF1 C-terminal fragment (UPF1-C-ter) encompassing the helicase core was previously reported to partially restore NMD after deletion of the *UPF1* gene[39]. We took advantage of our functional UPF1$^{AKS \rightarrow HPA}$ mutant and designed wild-type and mutant UPF1 expression vectors under doxycycline repressible tetO7 promoter, which we used to transform *S. cerevisiae upf1Δ* strains. We first assessed UPF1 protein levels in wild-type and mutant strains to verify that the mutation does not alter the protein expression (Fig. 5a). We then assessed NMD efficiency in both contexts. To do so, we performed qPCRs on RNA extracts and quantified the

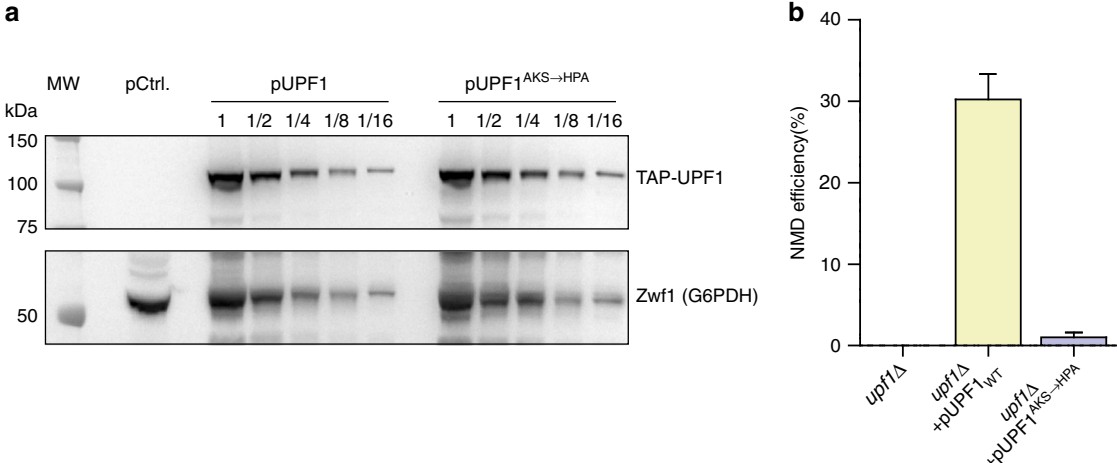

**Fig. 5** Loss of UPF1 processivity reduces NMD efficiency. **a** The levels of the expressed proteins, as tested with antibodies against the N-terminal TAP tag, were similar for UPF1 and UPF1[AKS→HPA], showing that the reduction in NMD efficiency is due to the mutation and not to a reduction in UPF1 expression levels. Two-fold dilutions of protein extracts were separated on 4–12% polyacrylamide gels, transferred and probed. The signal for an abundant protein, Zwf1, was used as loading control. **b** To test the ability of the UPF1[AKS→HPA] variant to function in vivo in NMD, we used RT-QPCR to measure the steady-state levels of a natural NMD substrate, the transcript for the DAL7 gene. A upf1Δ strain was transformed with a control plasmid (pRS316), a plasmid expressing a region encompassing UPF1 helicase domain, and a plasmid expressing a mutant version (UPF1[AKS→HPA]) and compared with a wild-type (WT) strain transformed with the control plasmid. The changes in DAL7 RNA levels compared with wild-type were measured in three independent experiments and used to calculate NMD efficiency, with average and standard deviation values shown as a bar plot

expression level of DAL7, a natural transcript targeted by yeast NMD[40]. Overexpression of a wild-type UPF1-C-ter allowed recovery of a 30% efficient NMD in a yeast upf1Δ strain. This partial complementation effect was abolished when we used a mutant UPF1[AKS→HPA] version (Fig. 5b), suggesting that under limiting conditions, UPF1 tight binding and/or translocation processivity are required for NMD.

## Discussion

The human genome codes for more than 100 helicases implicated in multiple steps of both DNA and RNA metabolisms[3,41]. Each of them is involved in a specific pathway, and many are related to variable diseases, from cancers to developmental defects and neurodegenerative disorders[42–44]. Even though similarities in their helicase cores point towards a common ancestor, it has become evident that every helicase is unique and possesses structural elements leading to several layers of complexity.

UPF1-like helicases show a large functional diversity, with members involved in a variety of RNA regulation pathways. Their helicase core domains share a similar structural organization formed of two RecA-like domains and two extra domains protruding from the RecA domain 1A[17–20,33,45] that are flanked by different and specific N-terminal and C-terminal accessory domains. One could imagine that helicase cores from the same subfamily have conserved comparable intrinsic properties. Unexpectedly, our study shows that UPF1 and IGHMBP2 cores present contrasting helicase activities. IGHMBP2 is not capable of unwinding on its own and requires assistance of its C-terminal domains to unwind dsDNA with a very low processivity estimated to 19 bp, in clear contrast with the processivity of UPF1 estimated to more than 10 kb. We questioned the role of protrusions 1B and 1C in this disparity as the deletion of domains 1C or 1B leads respectively to a loss of UPF1 NA binding, or to uncoupling between NA binding and ATP hydrolysis in vitro, both abolishing NMD in vivo[17]. By simultaneously swapping 1B and 1C domains between UPF1 and IGHMBP2, we reveal that these protruding domains most likely co-evolved to dictate the

processivity of the helicase core to which they are attached. In the case of UPF1, domain 1B moves away from the RecA surface and towards 1C, possibly to form a circular channel for NA[17,33]. In the case of IGHMPB2, domain 1B is rather distorted and collapsed on bound NA[19] potentially disturbing the unwinding mechanism. Structural data of both parental and chimera helicases bound to NA would be needed to precisely understand how domains 1B and 1C contribute to the formation of the clip around the NA substrate.

The impact of protruding domains on the helicase processivity has been previously observed for SF1-A helicases for which the interaction between the domains 1B and 2B that protrude from each RecA domain directly alters the movement of the helicase core[1,30]. Recent structures of several UPF1-like helicases put forward the variability of protruding domains between these related enzymes. The nuclease/helicase DNA2 presents a C-terminal domain highly resembling to IGHMBP2 helicase[20]. Sen1 presents a shorter protrusion 1C, called the prong, and a supplementary brace that restricts the movement of protrusion 1B[18]. In the splicing factor Aquarius/IBP160, a domain called the pointer replaces the protrusion 1C at the exact same position and is probably responsible for the reversed 3′–5′ polarity of this helicase[45]. Using comparable methods, it will be particularly interesting to explore the mechanical variety of these motors and the role of protruding domains.

Mechanistic insights into the remarkable processivity of UPF1 came from the study of a large series of mutants that gradually affect its grip tightness. In addition to protruding domains swapping, we targeted a flexible loop at the hinge between 1C and the RecA domain 1A. This loop drew our attention due to its position outside any conserved helicase motif, its contribution to NA binding, its divergence among UPF1-like helicases, and its conformational changes upon NA binding[17,33]. We assessed our mutants with a combination of two single-molecule assays: (i) an unwinding assay in the presence of ATP, to measure helicase processivity in distance and in time; and (ii) a binding assay in absence of ATP, to measure the residence time of the enzyme once bound to NA as well as its sliding speed against the opposing force of hairpin closure. The sliding phenomenon shows that the tested helicases are comparable to

molecular binders with C-shaped spring clips that exist in three different states: a ground closed binding state with an initial binding energy, a partially opened sliding state and a fully opened unbinding state. For each enzyme, the transition from the ground state to the sliding or to the unbinding state follows an Arrhenius law since it depends on the energy fluctuation needed to alter the binding energy and open the clip: The stronger the initial grip, the higher is the energy fluctuation required to slide or fall, the more time is needed for the event to occur. In fact, the total residence time we measured for each of our enzymes is none-other than the time it takes for the clip to open sufficiently and fall, while the sliding time per base corresponds to smaller events where the clip opens just enough to slide by one base. For instance, wild-type UPF1 has a very tight grip on NA, and a very high binding energy in the ground state. Thus, a large energy fluctuation is required to lead to sliding or unbinding, and such events happen with an extremely low frequency.

Furthermore, in this clip model, two parameters govern binding to NA: the NA binding pocket constituted by a specific set of amino acids, and the inner stiffness of the clip, arising from the overall helicase structure. We suggest that the total binding energy ($E_{Bi}$) governing the clip state is the sum of the interaction energy ($E_{Int}$) inside the cavity of the clip, and the mechanical energy ($E_M$) stored in the spring of the clip when it is stretched. The differences in unbinding rates we recorded between the tested enzymes led us to calculate $E_{Bi}$ differences between the mutants. Using these values, we tested two possible models to see which of the two energies $E_{Int}$ and $E_M$ is altered by the mutations. Our data well fit a model in which mutations negatively alter $E_{Int}$ and not $E_M$ (see Supplementary Note) suggesting that all UPF1 variants tested in this study have roughly the same spring stiffness conferred by the protein scaffold, but diverge in their binding interactions with NA inside the clip.

Our clip model also explains the impact of the grip on helicase processivity. During ATP hydrolysis, power strokes drive consecutive conformational changes to move forward, but weaken helicase binding. Less energy is therefore required to open the clip, making it easier to fall during translocation, especially for enzymes with a weaker grip.

Finally, previous modeling of helicase processivity has considered the impact of NA sequence[31] and the destabilizing effect of the NA fork pushing the helicase[38], but no attempt has been made so far to look at helicase binding lifetime and its impact on processivity. Our model demonstrates that a helicase grip dictates its binding lifetime, which in turn limits its processivity. Remarkably, both our data and our model reveal that a single-residue mutation outside the conserved helicase motifs, that slightly reduces the binding energy of the helicase, is sufficient to drastically reduce its processivity. Lastly, we previously showed that UPF1 is an efficient NA–protein interaction remodeler that can notably melt biotin–streptavidin interaction[27]. We expect that the strong grip of UPF1 is probably necessary to displace stable NA-binding proteins.

The notion of helicase processivity is intuitively associated to the distance traveled by the enzyme. Progression over long distances is clearly essential for processes like genome replication for which important portions of dsDNA must be unwound to allow duplication. Such events also involve fast helicases as replication is under time constraint to be achieved before cell division. In contrast, in multiple cases, DNA and RNA helicases only need to translocate over short distances to unwind short dsNA regions, to move over a short distance or to remodel proximal NA–protein interactions. Does this imply that most helicases are not processive? Here, the thorough study of UPF1 helicase core attributes revealed that its processivity coincides with the enzyme ability to remain bound to its substrate for long periods of time without detaching. We assume that the residence time of a helicase onto NA is an important parameter for its

function, notably to offer a time window long enough to guarantee process completion. The process of NMD requires several successive steps including translation-dependent recognition of a premature termination codon, ribosome dissociation, mRNP remodeling, and recruitment of RNA decay factors[25]. In addition to the observation that ATP-dependent activities of UPF1 are essential for NMD, several evidences showed that UPF1 is involved in every successive step of NMD[25,46,47]. So, it has been proposed that ATP-dependent activities of UPF1 orchestrate the conformational and compositional transitions between PTC recognition and mRNA decay[25,48,49]. Here, the observation that UPF1 altered grip reduces NMD efficiency in yeast also argues for the notion that the residence time of UPF1 onto its substrate is an important parameter for NMD completion. The importance of UPF1 processivity is further supported by the observation that during HTLV infection in human cells, the viral protein Tax directly targets UPF1 translocation and reduces its processivity to decrease NMD efficiency[50]. Some helicases, like the SF2 RNA helicase eIF4A3, are deposited onto NA to form a stable and sequence-independent clamp without necessity to use ATPase activities[51,52]. In the case of UPF1, we suggest that it combines its tight grip and its ability to translocate both to serve as a binding platform for NMD factors and to remodel RNA–proteins complexes. Future investigations will be necessary to determine the precise site of action of UPF1 during NMD and whether NMD partners modulate its residence time and its processivity. Exploration of the biophysical attributes of closely related UPF1-like helicases will be necessary to better understand their action in vivo and the consequences of their mutations linked to several human disorders[13,14,53].

## Methods

**cDNA cloning and protein purification**. We produced helicase domains of *Homo sapiens* UPF1 (295-914), *Saccharomyces cerevisiae* UPF1 (221–851) and *Homo Sapiens* IGHMPB2 (1–652), as well as the full-length form of IGHMPB2 (1–993, IGHMPB2-FL) (Uniprot accession codes Q92900-2, P30771, and P38935, respectively). Boundaries were defined according to previous structural studies[17,19,33]. IGHMPB2, IGHMPB2-FL, and yeast UPF1 coding sequences were PCR amplified using oligos HLH 2696/2697, HLH 2696/2698, and HLH 2705/2706, respectively. Purified PCR fragments were inserted in a home-made variant of pET28a plasmid pHL5 (Novagen) between *Nde*I/*Not*I (IGHMPB2) and *Nhe*I/*Xho*I (yUPF1) restriction sites. To generate UPF1 mutants, we modified the wild-type coding sequence of plasmid pHL 1281 (yeast UPF1 helicase domain) through PCRs followed by ligation reactions using amplification oligos carrying the corresponding mutations at the targeted sites. Chimeric forms of UPF1 and IGHMPB2 were engineered using Gibson cloning strategy. Required fragments were amplified from yUPF1 (pHL 1281) and IGHMPB2 (pHL 1278). Domain frontiers selected for the swaps are indicated in Supplementary Figure 8A. After amplification and purification, the fragments were ligated through Gibson reactions. All recombinant proteins were fused to a CBP-tag at their N terminus and a hexa-histidine tag at their C terminus, and expressed using *Escherichia coli* BL21 competent strains (DE3) grown in LB medium and induced overnight at 16 °C. Cells were harvested and lysed in buffer A [1.5× PBS pH 7.5, 225 mM NaCl, 1 mM magnesium acetate, 0.1% (w/v) NP-40, 20 mM imidazole, 10% (w/v) glycerol] Supplemented with 100 mg ml⁻¹ of egg white lysozyme (Sigma-Aldrich) and 1× protease inhibitor cocktail EDTA-Free (Sigma-Aldrich). Proteins were first purified on Nickel columns (Ni-NTA, Qiagen) and further purified on a calmodulin affinity column. Collected proteins were dialyzed against 1.5× PBS, 10% (w/v) glycerol, 1 mM magnesium diacetate and 2 mM DTT, then stored at −80 °C.

**ATP hydrolysis assays**. ATP hydrolysis was performed in steady-state conditions. Proteins (10 pmol) were incubated at 30 °C in a 20 μl reaction mixture containing 1× ATPase buffer [20 mM MES pH 6.0, 100 mM potassium acetate, 1 mM DTT, 0.1 mM EDTA, 1 mM magnesium acetate, 1 mM zinc sulfate, and 5% (v/v) glycerol], 2 μCi of [γ32P]-ATP (3000 Ci/mmol, Perkin Elmer), 25 μM ATP and 50 μM polyC (concentration of binding sites). Reaction aliquots (2 μl) were withdrawn at various times and quenched with a buffer (5 μl) containing 10 mM EDTA and 0.5% (v/v) SDS. Samples were separated using thin layer chromatography on polyethyleneimine cellulose plates (Merck) with a 0.35 M potassium phosphate (pH 7.5) solvent, then analyzed using a Typhoon Phosphorimaging system. Quantification was performed using Fiji/ImageJ analysis package.

**Mutant UPF1 yeast expression vectors**. UPF1 wild-type and the mutant UPF1^(AKS→HPA) variant expressions in yeast were done with plasmids based on

pCM189-NTAP a single copy vector derived from pCM189 with a doxycycline repressible tetO7 promoter and an N-terminal TAP tag, an intermediate in the construction of pTG189. We amplified the 208–971 region of *UPF1* coding sequence with oligonucleotides CS1362 (TTAAGAAAATCTCATCCTCCGGGGC ACTTGATGCGAATAAAGACGCTACAATTAATGATATTGACG) and CS1364 (ATAACTAATTACATGATGCGGCCCTCCTGCAGGGCTTATATTCCCAAAT TGCTGAAGTC), having 35 nucleotides extremities identical with regions in the destination vector. *Not*I digested pCM189-TAP was used in a hot-fusion in vitro version of Gibson assembly followed by direct transformation in *E. coli*. To generate the AKS484–486HPA mutant, three partially overlapping PCR fragments were obtained on the *UPF1* plasmid template and on pHL1376. All the constructs were verified by Sanger sequencing. The plasmids were transformed in yeast wild-type and LMA1667 (*upf1Δ*) for RNA quantitation and immunoblotting.

**Yeast RNA quantitation**. Total RNA was extracted from cells grown to mid-exponential phase in rich medium (YPD, yeast extract, peptone, glucose) using the hot acid phenol protocol. DNAse-treated samples of equal total RNA amounts were reverse transcribed using specific oligonucleotides (CS888— CTCAGTTTGCGATGGAAGAG, CS1430—TCCCAACGACCACAGTTCAA ACC and CS1077—AACCGTCGTCTCTCTCGAAG). Q-PCR estimation of initial RNA amounts were done on eightfold dilutions of the RT reaction using oligo-nucleotides CS1429 (TGAAACTTTGCCAGCGGCCTTC)/CS1430 (DAL7) and CS1076 (GTTAGAAAAGGCGCTTTGGTATATG)/CS1077 (RIM1). We used the amount of RIM1 mRNA for normalization and used differences in Ct to estimate RNA fold change in comparison with the wild-type strain, as reference. Each experiment was performed at least three times.

**Yeast immunoblots**. Protein extracts were obtained by boiling equal numbers of mid-exponential growth cells in denaturing electrophoresis sample buffer. Two-fold dilutions of protein extracts were separated on 4–12% polyacrylamide gels (Novex NuPAGE MOPS running buffer, Life Technologies). The proteins were transferred by electro-blotting on nitrocellulose membranes and probed with PAP (peroxidase-anti-perox-idase antibodies, Sigma P1291, 1:2000), followed by a control test with polyclonal anti-Zwf1 (glucose-6-phosphate dehydrogenase) antibodies (Sigma A9521, 1:20,000). Per-oxidase activity was monitored with the Clarity Western ECL chemiluminescence kit (Bio-Rad) and pictured on a Bio-Rad GelDoc imaging system. Image processing was performed with FIJI/ImageJ and consisted in contrast adjustment. Uncropped blots are presented in Supplementary Figure 8B.

**Single-molecule data acquisition**. Camera images were used to collect the raw data of DNA extension. Raw data represented the real-time evolution of the DNA extension in nm, which was converted into the number of base pairs unzipped by helicases using a calibration factor determined from the elastic properties of ssDNA (Supplementary Figure 8C). Instantaneous unwinding and translocation rates were obtained from a linear fit to the traces filtered with a least square method over a time window define by the user mouse. The histograms of the instantaneous rates were fit to Gaussian functions when applicable. The error bars shown in the histograms are proportional to the inverse of the square root of the number of points for each individual bin.

**Experiment with magnetic tweezers**. The substrate used in single-molecule assays is a 1239 bp long DNA hairpin that has a 76-nt 5′-biotinylated ssDNA tail and a 146 bp 3′-digoxigenin-labeled dsDNA tail[27,34]. The biotinylated end of the hairpin is attached to a streptavidin coated paramagnetic bead while the digox-igenin end is attached to the surface of a flow cell via anti-digoxigenin. We used a picotwist magnetic tweezers to manipulate the DNA substrate. A tunable force can be applied to the substrate by applying a magnetic field on the paramagnetic bead via a pair of permanent magnets. Force applied on the hairpin can be precisely controlled by changing the position of the magnet over the substrate. DNA extension was measured in nanometers in real time via a 31 Hz video-camera (CM-140 GE Jai) and then converted to base pairs by exploiting the elastic properties of ssDNA. All the measurements were conducted at 30 °C, unless mentioned other-wise. The working buffer had a composition of 20 mM Tris-HCl, 75 mM potassium acetate, 3 mM magnesium chloride, 1% BSA, 1 mM DTT, and 2 mM ATP. The concentration of helicase used was the lowest possible to observe single-molecule events and varied between 1 and 20 nM.

**Single-molecule binding assays (SMBA)**. Binding of proteins to DNA in single-molecule configuration was measured in the working buffer in absence of ATP, using cycles of hairpin openings and closings. Briefly, after helicase injection in a helicase buffer free of ATP, a cyclic variation of force is executed. Starting from low force where the hairpin is closed, the force is first increased for 1 s to the test force = 7 pN, to check that the hairpin is in the closed state then the force is increased to >15 pN for 3 to 4 s (Fig. 2, step 1) then reduced to 7 pN and held for 20 to 30 s (Fig. 2, step 2). The force is finally reduced below 3 pN (Fig. 2, step 3). The whole process is repeated several times to form a cyclic assay. The 3 pN regime closes the hairpin in most cases, this offers the possibility to check that the hairpin does open and close at each cycle. Multiple openings and closings of the hairpins provide various instances for proteins to bind with most of the available substrates. Helicase

concentration used is the lowest possible to observe single-binding events and varied between 1 and 20 nM.

**Single-molecule data analysis**. For unwinding processivity, large processivity was determined by taking into account only those traces where helicase starts unwinding the hairpin in fully closed state until it falls off. This leads to a histogram with a few points corresponding to excursion shorter than the hairpin length and a last bin of this precise length with many events (Supplementary Figure 2C). This truncated histogram was fitted to a truncated exponential providing a processivity value usually larger than the hairpin length. For moderately processive helicase, the histogram of the length of unwound events was calculated. These histograms were exponential in nature, which gave the mean processivity of the helicase on fitting them. The errors in his-tograms are given by the square root of the number of events in each bin. For residence time measurement in ATP-free SMBA, binding time was obtained by measuring the inverse of the falling rate. That is by summing the time over which the fork remained blocked by the helicase and dividing it over the number of real unbinding events that we have recorded. As the binding time is often long, the experiment's recording was often stopped before the binding event finished. Such event contributes to the total binding time but not to the number of unbinding events. The relative error on the binding time is computed as the $N_U^{-1/2}$.

## Data availability
The data that support the findings of this study are available from the corresponding authors upon reasonable request. Single-molecule data supporting the findings of this study are available at http://pimprenelle.lps.ens.fr/Magnetic_tweezers_graphs_Kanaan.zip (2.2 Go). Data analysis software (PlayItAgainSam) can be downloaded at (http://www.picotwist.com/download/Pias_setup.exe). Figures 1b, c, 2c, d and 3a–d in the main text have associated raw data. Supplementary Figures 2a, d, 3a, 4a–f, 6a–f, 7a, b and 8c have associated raw data.

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

## Acknowledgements

We thank E. Conti and B. Séraphin for plasmids, I. Barbosa and M.H. Pontier for experimental help, F. Fiorini for scientific advice, O. Bensaude, A. Jacquier, N. Leulliot and D. Bensimon for reading the manuscript, and members of our labs for discussion. This study was supported by the ANR CLEANMD grant (ANR-14-CE10-0014) from the French Agence Nationale de la Recherche to C.S., V.C., and H.L.H., by the program « Investissements d'Avenir » launched by the French Government and implemented by ANR (ANR–10–LABX-54 MEMOLIFE and ANR–10–IDEX–0001–02 PSL* Research University) to J.K. and H.L.H., by the European Research Council grant Magreps [267 862] to V.C. and S.R., by the Fondation ARC pour la recherche sur le cancer PhD fellowship to J.K., and by continuous financial support from the Centre National de Recherche Scientifique, the Ecole Normale Supérieure and the Institut National de la Santé et de la Recherche Médicale, France.

## Author contributions

H.L.H. and V.C. conceived the project. J.K. realized the cloning and the mutagenesis, purified the recombinant proteins, and performed the biochemical assays. V.C. built the magnetic tweezers. S.R. conducted single-molecule assays and conceived single-molecule binding assay. S.R., J.K., H.L.H. and V.C. analyzed the data. L.D. and C.S. performed the functional assays in yeast and analyzed the data. J.K., S.R., V.C. and H.L.H. wrote the paper.

## Additional information



