## [Peer Review File · Nature Communications]

Reviewers' comments:

Reviewer #1 (Remarks to the Author):

In their manuscript "UPF1-like helicase grip on nucleic acids dictates processivity" Kanaan and coworkers describe single-molecule analysis of the unwinding processivity and ATP-free DNA unbinding rate of many specific mutants of two related helicases. This large, quantitative, and high-resolution data set acquired with specific mutations that affect processivity and binding affinity support an intriguing and possibly general model for helicase processivity. The model predicts that nucleic acid binding by the helicase is mediated by a "spring clip" mechanical association combined with a protein specific binding energy. Unbinding from the DNA corresponds to both overcoming the specific binding energy and the energy required to open the "clip" to permit unbinding. The extensive single-molecule data are reasonably well fit by a version of this model in which the mutations affect the binding energy but do not alter the stiffness of the mechanical spring.

This is a well-conceived and thorough systematic investigation of the effects of mutants on the processivity of a helicase (UPF1 and a homolog). Helicase processivity is an important property of this important class of enzymes, but general models relating protein features to helicase processivity have not been developed. The data are of sufficient quality and the mutations generate a large enough change in the processivity and unbinding rate for this helicase that quantitative modeling can be attempted. The model is relatively simple but it fits the data reasonably well and it makes testable predictions concerning other aspects of helicase activity. As the authors demonstrate with the *in vivo* measurements of the nonsense mediated decay that requires the helicase UPF1, the processivity of helicases varies widely and can significantly impact the physiological roles played by these helicases. Specifically, the authors demonstrate that reducing the processivity of UPF1 results in reduced nonsense mediated decay of a specific RNA transcript. The proposed model for processivity is therefore an important and possibly general result for the helicase field and as such will be of interest to a broad audience interested in helicases and the myriad crucial functions they play cellular processes. I support publication of this work in Nature Communications. Nonetheless I have a number of points for the authors to address prior to acceptance.

1. The implicit assumption seems to be made that the ATP free state is the weak binding state of the helicase. Formally this could be any of several nucleotide states of the helicase, which is addressed to some extent in the model section that discusses the processivity in relation to the binding affinity. To probe this point in detail, have the authors considered measuring the sliding and off rate of a single mutant or WT helicase for different nucleotide states? This would provide another route to calculating the processivity of the helicase during processive unwinding. This would also bolster the result that lifetime during unwinding is always shorter than the lifetime of the ATP-free helicase on DNA, which must imply a weaker bound state for some particular nucleotide state of the helicase. If this could be established for one helicase variant then it would provide additional support for the model and could permit a quantitative analysis of the ATP-free unbinding time versus the unbinding time during processive ATP-dependent unwinding.

2. The data seem broadly consistent with the model premise that T_u , T_s , and T_r are linearly related, this relationship is indirect in the two plots in which the times are plotted together (Fig 4 and Fig S7). It would be useful to see T_u and T_s plotted as a function of T_r in a supplemental figure.

3. Along the same lines, the data for the residence time and sliding time are narrowly distributed around the fit line in Figures 4 and S7, but the unwinding lifetime is much more variable. Can the authors comment on the source of this variability and if it can be explained within the context of the proposed model or if it suggests that possible extensions of the model need to be considered to

accurately represent the unbinding time during unwinding.

4. The proposed model is hopefully a general model for helicase processivity. Whereas the extensive data that the authors have collected with several mutants with altered processivity has not previously been collected, it would bolster the general applicability of the model if the authors could demonstrate that the model could explain some aspect of existing measurements with other helicases. For example, the processivity as a function of ATP that is predicted by the model could potentially be compared with existing data relating helicase processivity to ATP concentration.

5. The model predicts that the sliding state is in some sense an intermediate between bound and free. Depending on the relative energies the model seems to predict that there should be reversible transitions from tightly bound to sliding, and vice versa. Have these transitions from sliding to tightly bound been observed? Given the energies and timescales of the experiments should these reversible sliding events be observed? This analysis could help bolster the model since these rates should be calculable and if they can also be observed then it may provide an internal consistency check on the model.

6. On line 200 the authors refer to the spring stiffness but the underlying model explaining the stiffness is described in the SI. I suggest giving a brief explanation or rephrasing this sentence since the concept and relevance of the spring stiffness has not been established.

7. Fig 4. As mentioned above, the T_s and T_r data are well fit by the model whereas the T_u data seem much more variable. Can the authors comment on this and provide some explanation for this large variability of T_u compared to the other two measures?

8. Fig 5. The legend should include a description of the 2-fold dilutions in loading that is indicated but not explained on the figure.

9. Line 244 – this is a very confusing sentence.

10. Line 288 – the authors make the claim that the mutations alter the binding energy but do not perturb the spring constant of the helicases. It would be helpful if they could speculate as to protein elements or structures that may contribute to the stiffness.

11. Line 290 – as touched on previously, has the affinity in the presence of ATP analogs been determined for any of the constructs. Even an ensemble measure of affinity could be an interesting data set to include that could provide additional support for the processivity model.

12. The UPFF1 processivity is reported to be on the order of 10 kb. This seems excessive to carry out its role in NMD. Are the RNAs that are degraded through this mechanism in excess of 10 KB?

13. P21 – methods: “image manipulation” is not what the authors meant I think. “image processing” or “image analysis” would be a better expression.

14. It seems that the authors are arguing that the energy of ATP hydrolysis is so large that it can easily overcome the binding and spring slip energies, but I wonder if there should be any relationship between the binding energy and the velocity. This relationship may prove interesting either way – it may prove the point the authors are making if there is no correlation, or it may point to something else if there is a correlation. The authors have the data so it would be nice to see this relationship between the binding energy and the unwinding velocity included in the SI.

15. The unbinding assay is an elegant extension of similar approaches developed in the croquette lab.

For many helicases the affinity for a ssDNA-dsDNA junction is higher than for ssDNA. In the unbinding assay the hairpin could potentially trap helicases in two different orientations on the DNA. In one orientation the helicase would be bound with the closing hairpin facing it, in which case it would be binding a substrate that resembled a ssDNA-dsDNA junction. In the opposite orientation the helicase would have the hairpin closing "behind" it and it would effectively be bound to only ssDNA. Is the relative affinity of UPF1 for ssDNA versus an ssDNA-dsDNA junction known? Would the authors expect to see a difference in binding times in the two possible orientations? It might be worth mentioning this detail since this approach is general and could be applied to helicases with potentially measurable off rate differences depending on the binding orientation.

16. Sup fig 2. The legend for part B needs to be much better described and the assay should be somewhat better described. For example, is the fraction plotted equal to the fraction of total ATP? In part c it looks like there were three events that were not completely unwound rather than only two.

17. Sup fig 5. There are no E coli enzymes in the alignment.

18. Sup fig 7. This figure is very confusing. The parameters listed below each graph for parts C and D need to be better described. The green and dashed lines in part C need to be described. The rationale for the unusual units on the axis in part C should be provided. In part D the sliding time is defined as a rate – this should be better explained.

19. Given the predictions from fig S7 part D for the backsliding, should it be observed for any of the measurements? Was it observed in any of the measurements? Once again if there is evidence for backsliding during unwinding that is consistent with the model this would be important to demonstrate. Conversely, if the model predicts observable levels of backsliding that were not observed then this should be commented on.

Overall, I found this to be a well-conceived and well-executed study that obtained significant results and developed a novel model that could have implications for understanding helicase activity beyond the model system used for this study. The authors did a very good job of discussing the limitation and caveats of their model, which I applaud. The writing was serviceable though it could be improved slightly with an additional round of editing. Once the authors have addressed the points I describe above then I think that this work would be of significant interest to a wide audience and would spur additional theoretical and experimental work to test and extend the model presented.

Reviewer #2 (Remarks to the Author):

This manuscript examines an underappreciated determinant of the processivity of helicase-catalyzed nucleic acids unwinding, the binding interaction energy. They found, using magnetic tweezers, two related RNA helicases have markedly different processivity in DNA unwinding, one with over 10,000 bp and the other with only about 20 bp. Based on structural comparison, they made a number of mutations, some with single to triplet amino-acids substitutions, and others with large subdomain swaps. A large scale swap made the less processive core more processive and vice versa. Small scale mutations made the more processive enzyme, yUpf1, less processive to various degrees. They also developed a novel method to determine the lifetime of the bound state for the helicase in the absence of ATP, and showed that there is a nice correlation between the unwinding processivity and the bound state lifetime. They then extended a previous model on helicase mechanisms by including the bound state lifetime, and the in vivo data suggest that these findings are functionally relevant in NMD.

Overall, the findings are novel and the scope of the analysis is extensive. It should appeal to a broad

audience and I recommend publication. Below are a few comments that the authors might consider before finalizing the manuscript.

“Though the processivity of other UPF1-like helicases was not determined so far, the processivity of hUPF1 largely exceeds that of DNA helicases like the SF1A helicases UvrD or Rep determined by single molecule approaches”.

-This statement is confusing because they are comparing unwinding processivity determined here to unwinding processivity of an engineered Rep helicase to make it much more processive than the wild type. Rep-X in Arslan et al was shown to unwinding 4-5 kb of DNA processively without falling off whereas here only 1.2 kb of DNA unwinding was shown. So I don't think it is appropriate to say that hUpf1 has a higher processivity than Rep-X of Arslan et al. On the other hand, Rep or UvrD monomers have extremely low processivity and unwinding is shown only when two or more are bound in Lee et al for the case of UvrD. In Lee et al, unwinding processivity was not explicitly measured when two UvrD molecules were unwinding the DNA. The authors should make more explicit comparisons because the unwinding activities of wild type Rep/UvrD monomers, UvrD dimer and engineered Rep (Rep-X) are vastly different. In addition, Comstock et al showed that UvrD monomer does unwinding 10-20 bp under unzipping tension but when two monomers cooperative, a full hairpin can be unwound.

2. “surprising given that hUPF1 monomers translocate onto NA at least one order of magnitude slower”

-It is not clear to me why a slower translocate necessarily should be expected to be less processive in unwinding.

3. In Fig. S2, they say that unwinding processivity is 19 bp. But they only show only one time trace and do not show the relevant statistics such as a histogram of the number of unwound basepairs per event.

4. “60 seconds (with C119 ter, n=27) (fig. S3C)” This is not in figure s3c.

5. “They also reveal a functional coordination between auxiliary domains 1B and 1C”

I don't think the data reveals a functional coordination between the two subdomains. If there is, the authors should explain the reasoning behind the statement.

6. In supplementary text on the model, r_U , r_S , r_R must be tau's. Also, I feel that this section can be improved in terms of its clarity.

Reviewer #3 (Remarks to the Author):

The superfamily 1 helicase UPF1 is an important enzyme and its roles in NMD and telomere regulation are well documented. Structural information is available for several UPF1-like helicases but their mechanisms of action have not been probed in detail. The Croquette and Le Hir labs have addressed an important question concerning UPF1 translocation on a nucleic acid lattice. Although the ring-shaped nature of the hexameric helicases can go some way towards explaining their processivity, the monomeric SF1 and SF2 helicases can also display remarkable processivity but how this is achieved is unclear.

In this study the authors have used single molecule magnetic tweezers to measure

translocation/unwinding and the binding lifetimes of UPF family helicases on DNA. They identify specific structural elements inserted in the conserved helicase core of UPF1 that impart nucleic acid binding stability and processivity. Specific amino acids are identified that impact on UPF1-DNA interactions in vitro. One mutant allele of yeast UPF1 was tested in intact cells and demonstrated a defect in NMD as predicted. To the best of my knowledge these observations are novel and if correct they add to a growing body of information indicating that, within helicase family members, helicase core has evolved to provide specificity of function.

All mechanistic observations have been made in only one experimental system. That said, the properties of processivity and binding half-time measured for the wild-type and a large number of variant UPF forms adhere well to the proposed "clip" model. Their case would be stronger if more mutants were tested in vivo and the results extended to human UPF1.

The manuscript is clearly written and easy to follow, but there is a need for some minor English language editing. Some suggestions to improve clarity are as follows:

(1) Supplementary Figure 1 could be improved by the addition of the IGHMBP2 sequence and the indication of relevant domains, in particular protrusions 1B and 1C and their territorial limits. There is of course some overlap with Supp. Figure 5, where the text is quite small and I assume that the third line "SMBP2" is IGHMBP2.

(2) The mutated residues AKSR are not indicated as part of protrusion 1C in Supp. figure 5. It is not entirely clear to me whether these residues are regarded as part of the protrusion 1C, the RecA fold, or indeed a separate entity between the two. They seem to be an integral feature of the ssDNA binding channel however. Was this short 4 residue segment included in the original domain swap? The amino acid coordinates of the swapped domains are buried in Supp. Fig 8A. I would prefer to see them quoted in the main manuscript with reference to, perhaps, Supp. Fig 1. I have to hunt around a bit in all the information to piece together an answer to this.

(3) Although I agree that the data for the domain swaps indicate a role for domains 1B and 1C in helicase "grip" and processivity, I am not clear on how they reveal a "functional coordination" between the two auxiliary domains (line 150, P7). Were any critical residues in protrusions 1B and 1C identified and tested?

(4) Did I miss a table? The names of cloning/mutagenesis/Q-PCR oligos are not useful when their sequence are not given. Anyway, are they really necessary in the case of the cloning and mutagenesis?

(5) Concerning the in vivo experiments, Line 216 of the results states that "overexpression of a wild type UPF1-Cter allowed recovery of a 30% efficient NMD...". The band in the Fig. 5A appears to be the size of full-length UPF1 (but I don't know how big the tag is) and the materials and methods described production of full-length constructs, I think. So, it is not clear to me what was used.

Responses to Reviewer's comments

We would like to thank the Reviewers for their careful evaluation of the manuscript and their numerous comments. In response, we have included new experiments, new analyses and made several modifications to figures as well as clarified the text (all changes are indicated in red in the main text). We notably described our model in more details in supplementary information. We describe below our answers to the comments of the Reviewers. We feel that these changes have greatly improved the manuscript.

Reviewer #1 (Remarks to the Author)

In their manuscript "UPF1-like helicase grip on nucleic acids dictates processivity" Kanaan and coworkers describe single-molecule analysis of the unwinding processivity and ATP-free DNA unbinding rate of many specific mutants of two related helicases. This large, quantitative, and high-resolution data set acquired with specific mutations that affect processivity and binding affinity support an intriguing and possibly general model for helicase processivity. The model predicts that nucleic acid binding by the helicase is mediated by a "spring clip" mechanical association combined with a protein specific binding energy. Unbinding from the DNA corresponds to both overcoming the specific binding energy and the energy required to open the "clip" to permit unbinding. The extensive single-molecule data are reasonably well fit by a version of this model in which the mutations affect the binding energy but do not alter the stiffness of the mechanical spring.

This is a well-conceived and thorough systematic investigation of the effects of mutants on the processivity of a helicase (UPF1 and a homolog). Helicase processivity is an important property of this important class of enzymes, but general models relating protein features to helicase processivity have not been developed. The data are of sufficient quality and the mutations generate a large enough change in the processivity and unbinding rate for this helicase that quantitative modeling can be attempted. The model is relatively simple but it fits the data reasonably well and it makes testable predictions concerning other aspects of helicase activity. As the authors demonstrate with the *in vivo* measurements of the nonsense mediated decay that requires the helicase UPF1, the processivity of helicases varies widely and can significantly impact the physiological roles played by these helicases. Specifically, the authors demonstrate that reducing the processivity of UPF1 results in reduced nonsense mediated decay of a specific RNA transcript. The proposed model for processivity is therefore an important and possibly general result for the helicase field and as such will be of interest to a broad audience interested in helicases and the myriad crucial functions they play cellular processes. I support publication of this work in Nature Communications. Nonetheless I have a number of points for the authors to address prior to acceptance.

1. The implicit assumption seems to be made that the ATP-free state is the weak binding state of the helicase. Formally this could be any of several nucleotide states of the helicase, which is addressed to some extent in the model section that discusses the processivity in relation to the binding affinity. To probe this point in detail, have the authors considered measuring the sliding and off rate of a single mutant or WT helicase for different nucleotide states? This would provide another route to calculating the processivity of the helicase during processive unwinding. This would also bolster the result that lifetime during unwinding is always shorter than the lifetime of the ATP-free helicase on DNA, which must imply a weaker bound state for some particular nucleotide state of the helicase. If this could be established for one helicase variant then it would provide additional support for the model and

could permit a quantitative analysis of the ATP-free unbinding time versus the unbinding time during processive ATP-dependent unwinding.

We agree with the reviewer that any step or even several steps of the ATP hydrolysis cycle could critically jeopardize helicase binding. In order to probe this point, we selected one UPF1 mutant (γ UPF1 R487S) and assessed its binding in presence of ADPNP, a non-hydrolysable ATP analog. We evaluated helicase binding in these conditions using our single molecule binding assay, and measured the total residence time (τ_R) and the sliding slope in presence of the analog. We recorded 58 binding events, with an average binding time τ_R of 1631 ± 200 seconds and an average sliding slope of -0.109 ± 0.01 nm/s. Both values are smaller than the measures in total absence of ATP that we report in our manuscript (respectively 1845 ± 406 s and -0.063 nm/s) indicating that the binding lifetime is reduced in the ATP bound state.

We have now added this result in the revised main text as follows:

“To test this hypothesis, we chose the moderately affected mutant UPF1R487→S and tested by SMBA the impact of ADPNP, a non-hydrolysable analogue of ATP. Interestingly, both the binding lifetime and the sliding slope of this mutant were significantly reduced in the presence of ADPNP (Table S1).”

An extensive study of all the proteins figuring in this manuscript as well as helicases from other families in presence of different ATP analogs mimicking different transition states will be extremely valuable to explore the mechanism of ATP hydrolysis and its impact on helicase binding. However, we consider that such a study would be beyond the scope of this manuscript.

2. The data seem broadly consistent with the model premise that τ_U , τ_S , and τ_R are linearly related, this relationship is indirect in the two plots in which the times are plotted together (Fig 4 and Fig S7). It would be useful to see τ_U and τ_S plotted as a function of τ_R in a supplemental figure.

We thank the reviewer for his advice. For more clarity, we have now added two plots showing τ_U and τ_S plotted as a function of τ_R as supplementary figures S7A and S7B.

3. Along the same lines, the data for the residence time and sliding time are narrowly distributed around the fit line in Figures 4 and S7, but the unwinding lifetime is much more variable. Can the authors comment on the source of this variability and if it can be explained within the context of the proposed model or if it suggests that possible extensions of the model need to be considered to accurately represent the unbinding time during unwinding.

We fully agree with the reviewer that the distribution of unwinding lifetimes shows higher variability around the fit line.

Within the context of the model, this variability may be understood as follows. When we look at the relation describing τ_R and τ_S , they follow the Arrhenius law and we can write:

$$\tau_{Ri} = (1/\gamma_0) \cdot \exp(E_{bi} + k_G \cdot x_R^2/2)/k_B T = r_R \cdot \exp(E_{bi} - E_{b1})/k_B T$$

$$\tau_{Si} = (1/\gamma_0) \cdot \exp(E_{bi} + k_G \cdot x_S^2/2)/k_B T = r_s \cdot \exp(E_{bi} - E_{b1})/k_B T$$

with

$$r_R = (1/\gamma_0) \cdot \exp(E_{b1} + k_G \cdot x_R^2/2)/k_B T$$

$$r_s = (1/\gamma_0) \cdot \exp(E_{b1} + k_G \cdot x_S^2/2)/k_B T$$

In the case of τ_{Ui} you find the expression below, leading to:

$$\tau_{Ui} = (1/\gamma_0) \cdot (1/y_p) \cdot \exp(E_{bi} - \Delta E_p + k_G \cdot x_R^2/2)/k_B T = r_u \cdot \exp(E_{bi} - E_{b1})/k_B T$$

$$\text{with } r_u = (1/\gamma_0) \cdot (1/y_p) \cdot \exp(E_{b1} - \Delta E_p + k_G \cdot x_R^2/2)/k_B T$$

What is clear is that τ_{Ui} involves the extra parameter y_p compared with τ_R and τ_S . In our model, this extra parameter y_p represents the fraction of time the helicase spends in the open state induced by ATP hydrolysis. It is likely that this parameter varies slightly from one mutant to the next, making τ_{Ui} more variable.

4. The proposed model is hopefully a general model for helicase processivity. Whereas the extensive data that the authors have collected with several mutants with altered processivity has not previously been collected, it would bolster the general applicability of the model if the authors could demonstrate that the model could explain some aspect of existing measurements with other helicases. For example, the processivity as a function of ATP that is predicted by the model could potentially be compared with existing data relating helicase processivity to ATP concentration.

We thank the reviewer for this suggestion. So far, we have only found one enzyme for which the processivity and the rate were measured as a function of [ATP], that is the RecBCD enzyme.

In the work of Lucius and Lohman (J Mol Biol. 2004), the authors measured RecBCD rate change against the concentration of ATP, leading to a Michaelis-Menten constant K_M of $176 \pm 30 \mu\text{M}$.

In the work of Roman et al., RecBCD processivity was measured against ATP concentration, leading to a reaction constant of $41 \pm 9 \mu\text{M}$. This value corresponds to the reduced Michaelis-Menten constant $K_{M'}$ in our model (equation (10), supplementary information). The smallest value found corresponds to our prediction that when decreasing ATP, the helicase goes slower but as less chance to fall (“qui va piano va sano”).

$$\text{Based on our model: } K_{M'} = K_M \cdot \frac{\tau_U}{\tau_R}$$

Thus, using the values extracted from both publications, and within the restriction that these measurements were not done in exactly the same conditions, we can calculate the ratio $\frac{\tau_U}{\tau_R}$ belonging to [0,1].

$$\frac{\tau_U}{\tau_R} = \frac{K_{M'}}{K_M} = \frac{41}{176} = 0.23$$

As predicted in our model, $K_{M'}$ is smaller than K_M . These values also infer that $\tau_U/\tau_R = 23\%$ meaning that the enzyme spends 23% of its time in the ATP open state.

Lucius AL, Lohman TM. Effects of temperature and ATP on the kinetic mechanism and kinetic step-size for E.coli RecBCD helicase-catalyzed DNA unwinding. J Mol Biol. 2004 Jun 11; 339(4):751-71.

Roman LJ, Eggleston AK, Kowalczykowski SC. Processivity of the DNA helicase activity of Escherichia coli RecBCD enzyme. J Biol Chem. 1992 Feb 25; 267(6):4207-14.

5. The model predicts that the sliding state is in some sense an intermediate between bound and free. Depending on the relative energies the model seems to predict that there should be reversible transitions from tightly bound to sliding, and vice versa. Have these transitions from sliding to tightly bound been observed? Given the energies and timescales of the experiments should these reversible sliding events be observed? This analysis could help bolster the model since these rates should be calculable and if they can also be observed then it may provide an internal consistency check on the model.

We indeed have tried to visualize the sliding signal steps that would correspond to these transitions but so far, our instrumental resolution (2 to 3 bp) is not sufficient to reach that goal (we need to distinguish single base steps). We do hope that instruments offering higher resolution will soon allow us to observe and consider these sliding events.

6. On line 200 the authors refer to the spring stiffness but the underlying model explaining the stiffness is described in the SI. I suggest giving a brief explanation or rephrasing this sentence since the concept and relevance of the spring stiffness has not been established.

For better clarity, we have now removed the reference to spring stiffness from the main text in the part entitled “*A novel mechanistic model links helicase grip, binding lifetime and processivity*” in our results section, and only refer to it in the discussion. We also edited the corresponding paragraph in the discussion to better describe the stiffness clip notion.

7. Fig 4. As mentioned above, the T_s and T_r data are well fit by the model whereas the T_u data seem much more variable. Can the authors comment on this and provide some explanation for this large variability of T_u compared to the other two measures?

We have responded to this question above at point 3.

8. Fig 5. The legend should include a description of the 2-fold dilutions in loading that is indicated but not explained on the figure.

We have now added the following sentence in the description of figure 5A: “*Two-fold dilutions of protein extracts were separated on 4-12% polyacrylamide gels, transferred and probed*”.

9. Line 244 – this is a very confusing sentence.

We apologize for the confusion. We have now rewritten this sentence for more clarity:

“*In order to understand how protrusions 1B and 1C contribute to RNA binding and processivity, it will be useful to determine the structures of both the parental and chimeric helicases described in this study, in presence of long ssNA and dsNA forks.*”

10. Line 288 – the authors make the claim that the mutations alter the binding energy but do not perturb the spring constant of the helicases. It would be helpful if they could speculate as to protein elements or structures that may contribute to the stiffness.

As we mentioned in our discussion, a deeper understanding of the clip mechanism would require supplementary structural data. For instance, crystal structures of the mutants generated in our study could illustrate the conformational changes leading to alteration in the binding energy. As for the stiffness, if we had to speculate, one probable contributor to the clip stiffness would be the RecA-like domains, as they are key players in the helicase motor function. Indeed we did not alter these domains themselves, nor the conserved helicase motifs or the junction between Rec1A and Rec2A. Together, these domains create the binding

surface to nucleic acids; since the amino acids involved in nucleic acid binding are split between Rec1A and Rec2A, one could imagine that increasing or decreasing the distance between them would alter the clip rigidity.

11. Line 290 – as touched on previously, has the affinity in the presence of ATP analogs been determined for any of the constructs. Even an ensemble measure of affinity could be an interesting data set to include that could provide additional support for the processivity model.

In this work, we did not perform any ensemble assays to evaluate affinity in presence of analogs, but we have now tested the binding of one mutant (R487S) in presence of two ATP analogs, as described in question 1. However, in line with this idea, the affinity of human UPF1 helicase domain (hUPF1) to nucleic acids has previously been assessed in presence of ATP analogs:

- In the work of Chamieh et al 2008, pull downs with biotinylated RNAs showed that the quantity of recombinant hUPF1 precipitated was lower in the presence of ADPNP than in its absence (Figure 2A). Furthermore, in RNase protection assays, hUPF1 is mixed with a labeled RNA before RNase treatment. In presence of ADPNP, an overall lower quantity of labeled RNA is protected by hUPF1, suggesting a lower affinity or a destabilization of binding in presence of ADPNP.
- In the work of Fiorini et al 2013, the addition of ADPNP weakened binding and led to a reduction of the overall hUPF1 affinity to a labeled RNA in electrophoretic mobility shift assays.

All these observations coincide with an overall altered binding of UPF1 to NA during ATP hydrolysis.

12. The UPF1 processivity is reported to be on the order of 10 kb. This seems excessive to carry out its role in NMD. Are the RNAs that are degraded through this mechanism in excess of 10 KB?

Miura and his coworkers estimated the average length of an mRNA in budding yeast to be around 1250 nucleotides (Miura et al. 2008, BMC genomics). In humans, median mRNA length is around 2787 bp based on the geneBase Database (Piovesan et al., 2016). Median 5'UTR, CDS and 3'UTR lengths are respectively 203, 1278 and 938 bp. These values are lower than 10kB, suggesting that many NMD substrates are not very lengthy, and supporting our hypothesis that the most interesting UPF1 feature is the time it spends on substrates, rather than the distance it can travel. However the two longest human mRNAs have lengths of 109 kb (TITIN) and 44 kb (MUC16), and the mRNA with the longest 3'UTR where UPF1 is expected to act during NMD is ZBTB20, with a length of 24.5 kb. Thus we cannot exclude that traveling on long distances might be useful on specific sets of long substrates. Another possibility is that traveling on long distances could be required in other pathways in which UPF1 is involved that are still poorly described.

13. P21 – methods: “image manipulation” is not what the authors meant I think “image processing” or “image analysis” would be a better expression.

We do agree, we did not alter the image but only changed the contrast for better visibility. We have now replaced the word “*manipulation*” with the word “*processing*”.

14. It seems that the authors are arguing that the energy of ATP hydrolysis is so large that it can easily overcome the binding and spring slip energies, but I wonder if there should be any relationship between the binding energy and the velocity. This relationship may prove interesting either way – it may prove the point the authors are making if there is no correlation, or it may point to something else if there is a correlation. The authors have the

data so it would be nice to see this relationship between the binding energy and the unwinding velocity included in the SI.

The fact that the mechanical energy stored in the spring is smaller than the energy released by hydrolyzing one ATP is a check that the model is realistic. But aside this simple argument we cannot address with the present model the unwinding rate of the enzyme. It is certainly a future direction of research to couple this binding model with one describing the ATP cycle (such as the Betterton Julicher model) so as to come closer to a helicase model.

15. The unbinding assay is an elegant extension of similar approaches developed in the croquette lab. For many helicases, the affinity for a ssDNA-dsDNA junction is higher than for ssDNA. In the unbinding assay the hairpin could potentially trap helicases in two different orientations on the DNA. In one orientation, the helicase would be bound with the closing hairpin facing it, in which case it would be binding a substrate that resembled a ssDNA-dsDNA junction. In the opposite orientation the helicase would have the hairpin closing “behind” it and it would effectively be bound to only ssDNA. Is the relative affinity of UPF1 for ssDNA versus an ssDNA-dsDNA junction known? Would the authors expect to see a difference in binding times in the two possible orientations? It might be worth mentioning this detail since this approach is general and could be applied to helicases with potentially measurable off rate differences depending on the binding orientation.

This is a very interesting observation. We have investigated this issue but it is very difficult to determine in the binding assay without ATP in which direction the helicase does block the fork. If we found a blocking time distribution with two different times it would suggest exactly what the Referee is looking for. However, this really requires extensive statistics which is not easy with Upf1. In the case where the binding assay is done with ATP, we can detect the direction of the helicase according to its dynamical behavior: if the helicase unwinds the hairpin (going up) the helicase is facing the fork, if it translocates (going down), the fork is actually pushing the helicase in the back. We have done this exact assay for the Rep helicase as the figure below shows. On the left graph (up & down) we display traces obtained by opening and closing the hairpin with Rep and ATP in the chamber. We have selected here traces of 3 beads which are interrupted by the hairpin closing by the helicase. One can see that just at the closing, on some occasions, the helicase starts unwinding, while in others it translocates. We have separated the two event types in the next two panels: in Up (middle graph) we have kept all the unwinding events and in Down (third graph) all the translocation events. For 47 events 18 were unwinding and 29 translocating, statistically it is not clear that translocation is really favored.

16. Sup fig 2. The legend for part B needs to be much better described and the assay should be somewhat better described. For example, is the fraction plotted equal to the fraction of total ATP?

We now amended the legend of supplementary figure 2B as follows: “*Graph showing the fraction of ATP hydrolyzed as a function of time by hUPF1, yUPF1 and IGHMBP2 under conditions of steady state turnover, wherein the ATP concentration is in excess compared to the protein concentration. Proteins were separately incubated with an excess of γ 32P-ATP. Aliquots were withdrawn at various time points and quenched with a stop buffer. Radioactive inorganic phosphate released over time during hydrolysis was separated from unreacted ATP on a thin layer chromatography. Membranes were analyzed using an imaging system and radioactive signals emitted by inorganic phosphate and unreacted ATP were quantified using Fiji/ImageJ.*”

In part C it looks like there were three events that were not completely unwound rather than only two.

We apologize for this mistake. Indeed there were three incomplete events, and 58 total unwinding events. However, this mistake does not alter the estimated value of unwinding processivity. We have now amended the legend of part C.

17. Sup fig 5. There are no E coli enzymes in the alignment.

We apologize for the mistake; indeed, we did not include any *E. coli* enzymes in the alignment. We have now amended supplementary figure 5 legend accordingly.

18. Sup fig 7. This figure is very confusing. The parameters listed below each graph for parts C and D need to be better described. The green and dashed lines in part C need to be described. The rationale for the unusual units on the axis in part C should be provided. In part D the sliding time is defined as a rate – this should be better explained.

We have now amended the figure legend explaining the green and the dashed lines in C. We have also explained in more detail the x unit, corrected the annotations and replaced the word “*rate*” by the word “*time*”.

19. Given the predictions from fig S7 part D for the backsliding, should it be observed for any of the measurements? Was it observed in any of the measurements? Once again if there is evidence for backsliding during unwinding that is consistent with the model this would be important to demonstrate. Conversely, if the model predicts observable levels of backsliding that were not observed then this should be commented on.

Observing backsliding in an unambiguous fashion requires to observe single base steps which is just out of reach in our apparatus (having 2 to 3 bp resolution). In a recent study [55] using nanopore, a significant level of backsliding varying from 60% to 1% (depending on the nucleotide nature) was measured for HEL308 helicase. Moreover this backsliding was greatly enhanced by ADP which corroborates our model prediction but this is

not	done	on	Upf1.
-----	------	----	-------

Overall, I found this to be a well-conceived and well-executed study that obtained significant results and developed a novel model that could have implications for understanding helicase activity beyond the model system used for this study. The authors did a very good job of discussing the limitation and caveats of their model, which I applaud. The writing was serviceable though it could be improved slightly with an additional round of editing. Once the authors have addressed the points I describe above then I think that this work would be of

significant interest to a wide audience and would spur additional theoretical and experimental work to test and extend the model presented.

Reviewer #2 (Remarks to the Author):

This manuscript examines an underappreciated determinant of the processivity of helicase-catalyzed nucleic acids unwinding, the binding interaction energy. They found, using magnetic tweezers, two related RNA helicases have markedly different processivity in DNA unwinding, one with over 10,000 bp and the other with only about 20 bp. Based on structural comparison, they made a number of mutations, some with single to triplet amino-acids substitutions, and others with large subdomain swaps. A large scale swap made the less processive core more processive and vice versa. Small scale mutations made the more processive enzyme, yUpf1, less processive to various degrees. They also developed a novel method to determine the lifetime of the bound state for the helicase in the absence of ATP, and showed that there is a nice correlation between the unwinding processivity and the bound state lifetime. They then extended a previous model on helicase mechanisms by including the bound state lifetime, and the in vivo data suggest that these findings are functionally relevant in NMD. Overall, the findings are novel and the scope of the analysis is extensive. It should appeal to a broad audience and I recommend publication. Below are a few comments that the authors might consider before finalizing the manuscript.

“Though the processivity of other UPF1-like helicases was not determined so far, the processivity of hUPF1 largely exceeds that of DNA helicases like the SF1A helicases UvrD or Rep determined by single molecule approaches”.

-This statement is confusing because they are comparing unwinding processivity determined here to unwinding processivity of an engineered Rep helicase to make it much more processive than the wild type. Rep-X in Arslan et al was shown to unwinding 4-5 kb of DNA processively without falling off whereas here only 1.2 kb of DNA unwinding was shown. So I don't think it is appropriate to say that hUpf1 has a higher processivity than Rep-X of Arslan et al. On the other hand, Rep or UvrD monomers have extremely low processivity and unwinding is shown only when two or more are bound in Lee et al for the case of UvrD. In Lee et al, unwinding processivity was not explicitly measured when two UvrD molecules were unwinding the DNA. The authors should make more explicit comparisons because the unwinding activities of wild type Rep/UvrD monomers, UvrD dimer and engineered Rep (Rep-X) are vastly different. In addition, Comstock et al showed that UvrD monomer does unwinding 10-20 bp under unzipping tension but when two monomers cooperative, a full hairpin can be unwound.

We apologize for the mistake in the reference, we agree with the reviewer that the processivity of engineered Rep in Arslan et al. exceeds that of UPF1. We have now modified the sentence in the text and added the reference to Comstock et al showing that UvrD monomers have a lower processivity when acting alone.

“Though the processivity of other UPF1-like helicases was not determined so far, the processivity of hUPF1 exceeds that of the DNA helicases UvrD or Rep in their monomeric state when determined by single molecule approaches (Comstock et al. 2015; Dessinges et al. 2004)”

2. “surprising given that hUPF1 monomers translocate onto NA at least one order of magnitude slower”

-It is not clear to me why a slower translocate necessarily should be expected to be less processive in unwinding.

The purpose of this sentence was to highlight the low speed of UPF1, which suggests that this helicase spends more time on the substrate to travel an equivalent distance as to fast helicases like Rep, UvrD and NS3. This long binding time could involve more probability of falling and jeopardize processivity. We have now amended the sentence into: “*Therefore, to cover similar distances, hUPF1 must stay a much longer time on its substrate.*”

3. In Fig. S2, they say that unwinding processivity is 19 bp. But they show only one time trace and do not show the relevant statistics such as a histogram of the number of unwound basepairs per event.

We have now added a histogram as figure S2C showing the distribution of 54 IGHMBP2-FL unwinding events. This distribution led to an average processivity of 19 ± 1.5 bp.

4. “60 seconds (with C-ter, n=27) (fig. S3C)” This is not in figure S3C.

We apologize for the mistake, indeed figure S2C only shows the binding events distribution of IGHMBP2, not those of IGHMBP2-FL. We have now amended the sentence as follows: “*IGHMBP2 residence times followed exponential distributions with a mean of 20 seconds (no C-ter, n=48) (fig. S3C) or 60 seconds (with C-ter, n=27).*”

5. “They also reveal a functional coordination between auxiliary domains 1B and 1C”

I don’t think the data reveals a functional coordination between the two subdomains. If there is, the authors should explain the reasoning behind the statement.

We do agree with the reviewer that this sentence is confusing, and it is not necessary. Therefore, we have now removed it from the text.

6. In supplementary text on the model, r_U , r_S , r_R must be tau’s. Also, I feel that this section can be improved in terms of its clarity.

We apologize for the lack of clarity, but in that section we are indeed describing r_U , r_S , and r_R which represent constant values that depend on the media conditions such as viscosity. We have now rewritten and clarified the model description, and notably the definition of r_U , r_S , and r_R .

Reviewer #3 (Remarks to the Author):

The superfamily 1 helicase UPF1 is an important enzyme and its roles in NMD and telomere regulation are well documented. Structural information is available for several UPF1-like helicases but their mechanisms of action have not been probed in detail. The Croquette and Le Hir labs have addressed an important question concerning UPF1 translocation on a nucleic acid lattice. Although the ring-shaped nature of the hexameric helicases can go some way towards explaining their processivity, the monomeric SF1 and SF2 helicases can also display remarkable processivity but how this is achieved is unclear.

In this study the authors have used single molecule magnetic tweezers to measure translocation/unwinding and the binding lifetimes of UPF family helicases on DNA. They identify specific structural elements inserted in the conserved helicase core of UPF1 that impart nucleic acid binding stability and processivity. Specific amino acids are identified that impact on UPF1-DNA interactions in vitro. One mutant allele of yeast UPF1 was tested in intact cells and demonstrated a defect in NMD as predicted. To the best of my knowledge these observations are novel and if correct they add to a growing body of information

indicating that, within helicase family members, helicase core has evolved to provide specificity of function.

All mechanistic observations have been made in only one experimental system. That said, the properties of processivity and binding half-time measured for the wild-type and a large number of variant UPF forms adhere well to the proposed “clip” model. Their case would be stronger if more mutants were tested *in vivo* and the results extended to human UPF1.

The manuscript is clearly written and easy to follow, but there is a need for some minor English language editing.

Some suggestions to improve clarity are as follows:

- (1) Supplementary Figure 1 could be improved by the addition of the IGHMBP2 sequence and the indication of relevant domains, in particular protrusions 1B and 1C and their territorial limits.

The purpose of supplementary figure 1 was to show the high sequence identity between human and yeast UPF1 homologs to justify our choice of pursuing experiments using yeast UPF1 recombinant protein. This is why we did not include IGHMBP2 in the alignment, which can be found in supplementary figure 5 as the reviewer mentioned. However, as the reviewer suggested, we have now added the frontiers of protrusions 1B and 1C on IGHMBP2 and UPF1 domain schemes in figures 1B and 1C.

There is of course some overlap with Supp. Figure 5, where the text is quite small and I assume that the third line “SMBP2” is IGHMBP2.

We apologize for the confusion, indeed SMBP2 is another annotation of IGHMBP2. For the sake of clarity, we have now replaced “SMBP2” by “IGHMBP2” in supplementary figure 5.

- (2) The mutated residues AKSR are not indicated as part of protrusion 1C in Supp. figure 5. It is not entirely clear to me whether these residues are regarded as part of the protrusion 1C, the RecA fold, or indeed a separate entity between the two. They seem to be an integral feature of the ssDNA binding channel however. Was this short 4 residue segment included in the original domain swap? The amino acid coordinates of the swapped domains are buried in Supp. Fig 8A. I would prefer to see them quoted in the main manuscript with reference to, perhaps, Supp. Fig 1. I have to hunt around a bit in all the information to piece together an answer to this.

Based on the domain limits defined when the structure of human UPF1 was first described (Cheng et al., 2007), the AKSR residues are within domain Rec1A. Residues AKS (UPF1) and HPA (IGHMBP2) were not included in the original domain swap, which took place after the common arginine residue (R). Protrusion 1C is located 6 amino acids downstream AKSR in UPF1 sequence (yUPF1 493-547), while it is located 4 amino-acids downstream the equivalent HPAR sequence in IGHMBP2 (IGHMBP2 275-346).

- (3) Although I agree that the data for the domain swaps indicate a role for domains 1B and 1C in helicase “grip” and processivity, I am not clear on how they reveal a “functional coordination” between the two auxiliary domains (line 150, P7). Were any critical residues in protrusions 1B and 1C identified and tested?

Like explained above (Question 5, Reviewer 2), we decided not to talk about this notion of coordination and to remove the corresponding sentence from the manuscript.

(4) Did I miss a table? The names of cloning/mutagenesis/Q-PCR oligos are not useful when their sequence are not given. Anyway, are they really necessary in the case of the cloning and mutagenesis?

We apologize for the mistake and thank the Reviewer for noticing the lack of mutagenesis oligo sequences. We have now removed the oligo numbers from supplementary figure S8A.

(5) Concerning the in vivo experiments, Line 216 of the results states that “overexpression of a wild type UPF1-Cter allowed recovery of a 30% efficient NMD...”. The band in the Fig. 5A appears to be the size of full-length UPF1 (but I don’t know how big the tag is) and the materials and methods described production of full-length constructs, I think. So, it is not clear to me what was used.

Once again we apologize for the missing information concerning the precise domain of Upf1 used in this experiment and thank the reviewer for noticing it. The signal in figure 5A corresponds to a truncated yeast UPF1 construct composed of the helicase and C-terminal domains (UPF1 208-971), fused to a Tap tag. The tag is 21 kD, and the UPF1 truncation is 85 kD, leading to a total of 106 kD. We have now amended the corresponding material and methods section.

REVIEWERS' COMMENTS:

Reviewer #1 (Remarks to the Author):

The Authors have addressed my concerns with the initial submission and I am pleased to recommend publication of this innovative and informative work in Nature Communications.

Reviewer #2 (Remarks to the Author):

I am satisfied with the revision and recommend publication.